# Preoperative Phase Angle as a Risk Indicator in Cardiac Surgery—A Prospective Observational Study

**DOI:** 10.3390/nu14122491

**Published:** 2022-06-16

**Authors:** Sylvia Ryz, Larissa Nixdorf, Jürgen Puchinger, Andrea Lassnigg, Dominik Wiedemann, Martin H. Bernardi

**Affiliations:** 1Division of Cardiac Thoracic Vascular Anaesthesia and Intensive Care Medicine, Medical University of Vienna, A-1090 Vienna, Austria; sylvia.ryz@meduniwien.ac.at (S.R.); juergen.puchinger@gmx.at (J.P.); andrea.lassnigg@meduniwien.ac.at (A.L.); 2Department of Surgery, Medical University of Vienna, A-1090 Vienna, Austria; larissa.nixdorf@meduniwien.ac.at; 3Department of Cardiac Surgery, Medical University of Vienna, A-1090 Vienna, Austria; dominik.wiedemann@meduniwien.ac.at

**Keywords:** bioelectrical impedance analysis, cardiac surgery, phase angle

## Abstract

Background: The phase angle (PhA) can be used for prognostic assessments in critically ill patients. This study describes the perioperative course of PhA and associated risk indicators in a cohort of elective cardiac surgical patients. Methods: The PhA was measured in 168 patients once daily until postoperative day (POD) seven. Patients were split into two groups depending on their median preoperative PhA and analyzed for several clinical outcomes; logistic regression models were used. Results: The PhA decreased from preoperative (6.1° ± 1.9°) to a nadir on POD 2 (3.5° ± 2.5°, mean difference −2.6° (95% CI, −3.0°; −2.1°; *p* < 0.0001)). Patients with lower preoperative PhA were older (71.0 ± 9.1 vs. 60.9 ± 12.0 years; *p* < 0.0001) and frailer (3.1 ± 1.3 vs. 2.3 ± 1.1; *p* < 0.0001), needed more fluids (8388 ± 3168 vs. 7417 ± 2459 mL, *p* = 0.0287), and stayed longer in the ICU (3.7 ± 4.5 vs. 2.6 ± 3.8 days, *p* = 0.0182). Preoperative PhA was independently influenced by frailty (OR 0.77; 95% CI 0.61; 0.98; *p* = 0.0344) and cardiac function (OR 1.85; 95%CI 1.07; 3.19; *p* = 0.028), whereas the postoperative PhA decline was independently influenced by higher fluid balances (OR 0.86; 95% CI 0.75; 0.99; *p* = 0.0371) and longer cardiopulmonary bypass times (OR 0.99; 95% CI 0.98; 0.99; *p* = 0.0344). Conclusion: Perioperative PhA measurement is an easy-to-use bedside method that may critically influence risk evaluation for the outcome of cardiac surgery patients.

## 1. Introduction

The condition of a body can be described using various methods. One is to look at the body composition derived from the different body compartments, mainly muscle and fat mass. Body composition is influenced by many different factors and can simultaneously provide an indication of the state of health, nutrition, and fitness or organ dysfunction, and predict the clinical course of disease [1,2,3,4,5,6,7].

Cardiological and cardiac surgery patients are often multimorbid, more frail, and have poorer overall health and nutritional status than healthy individuals or other patients, which are associated with worse outcomes [8,9,10,11,12,13].

A commonly used and well-established method for the non-invasive determination of body composition is bioelectrical impedance analysis (BIA) [3,9,14,15]. One of the parameters determined by this measurement is the phase angle (PhA). In response to the application of an external current, the PhA describes the resistance and reactance of the human body. The PhA provides information about the integrity of the cell membrane and the extent of fluid redistribution between intracellular and extracellular fluid spaces, reflecting the mass of the body cells [9,16]. It has already been shown that patients with cardiovascular disease have a lower PhA than healthy individuals. Similarly, a low PhA has been shown to be a predictor of poorer outcome and increased mortality in various patient collectives in general, and especially in cardiac surgery patients [17,18,19,20].

This study was conducted to describe the perioperative course of PhA in elective cardiac surgical patients and to identify whether the preoperative PhA, as assessed by BIA, influences the perioperative clinical course in these patients. Additionally, we wanted to identify which preoperative and intraoperative patient- or procedure-related risk indicators influence the PhA.

## 2. Materials and Methods

### 2.1. Population Characteristics

In this prospective observational single-center study, we included 200 elective cardiac surgical patients with planned cardiopulmonary bypass (CPB). The study was performed at the Division of Cardiac Thoracic Vascular Anesthesia and Intensive Care Medicine at Medical University of Vienna and conducted between 31 October 2016 and 11 August 2019.

We excluded patients younger than 18 years of age, patients who were pregnant, and patients with preoperative chronic renal failure on renal replacement therapy. Additionally, we excluded patients who underwent emergency surgery, transplant surgery, pulmonary thromboendarterectomy, and elective cardiac assist device implantation. Finally, patients who had not given written informed consent to participate in the study were excluded.

### 2.2. Procedure, Data and Sample Collection

We prospectively recorded preoperative patient data, comorbidities, surgery- and procedure-related factors, and postoperative data. The following risk indicators were recorded: age, sex, BMI, frailty scale [21], serum albumin, resistance, reactance, asthma, chronic obstructive pulmonary disease (COPD), insulin-dependent diabetes mellitus (IDDM), non-insulin-dependent diabetes mellitus (NIDDM), chronic kidney disease, history of cardiac decompensation, peripheral artery occlusive disease (PAOD), atrial fibrillation, stable or unstable angina pectoris, left ventricular ejection fraction (LVEF), coronary artery bypass graft (CABG), valve procedure, combined procedure (i.e., CABG+ valve procedure), other procedures on CPB, reoperation, duration of anesthesia and surgery, CPB and aortic cross-clamp (AoCC) time, erythrocyte (PRBC) and fresh frozen plasma (FFP) transfusion, use of fibrinogen and coagulation factors, cumulative fluid balances (i.e., [intravenous and CPB administered crystalloids + colloids + blood products − urine output − ultrafiltration), simplified acute physiology score (SAPS) 3, sequential organ failure assessment (SOFA) score and length of stay in an intensive care unit (ICU). Patients were admitted to the study the day before surgery, after giving informed consent. Patient data were prospectively recorded at the time of admission and followed up until hospital discharge or for a maximum of 7 days.

### 2.3. Bioelectrical Impedance Analysis

Whole-body bioimpedance analysis was performed using a phase-sensitive device operating at 800 microamperes at an operating frequency of 50 kHz (BIA 101 AKERN S.R.L., Florence, Italy). The device was calibrated every morning using the standard control circuit supplied by the manufacturer with a known impedance [resistance (R) = 380 ohm; reactance (Xc) = 47 ohm. The accuracy of the device was 1% for R and 2% for Xc. For the BIA measurement, each participant was supine with limbs slightly spread apart from the body. Low-impedance disposable tab electrodes (Bianostic AT; data input GmbH Germany) were placed on the right side at metacarpal and metatarsal sites of the right wrist and ankle. Measurements were performed preoperatively before the induction of anesthesia and once daily postoperatively until postoperative day (POD) 7 or hospital discharge. The measured BIA variables were resistance (R) and reactance (Xc) and the PhA was calculated using the formula Arctan (Xc/R) × 180°/π [22].

### 2.4. Statistical Analysis

The different demographic and baseline clinical data are expressed as the mean and standard deviation (SD) or median with interquartile range (IQR) for metric variables and absolute frequencies for categorical variables.

First, the perioperative course of the PhA was described using boxplots at each respective timepoint.

Second, to examine the effects of the preoperative PhA on the perioperative course, the patient population was divided into two groups: depending on the median preoperative PhA, patients below or equal to PhA 5.84° were assigned to group PhA_low_, and patients above PhA 5.84° were assigned to group PhA_high_. Differences between groups were analyzed with Student’s *t*-tests for normally distributed variables and Mann–Whitney U tests for non-normally distributed continuous variables. A paired *t*-test was used to compare dependent samples. The χ^2^ test was used to test categorical variables.

Third, the association of 13 preoperative risk indicators on the preoperative PhA was evaluated using a univariable logistic regression model. Odds ratios (ORs) with 95% CIs were used to quantify the effect of patient comorbidities. Next, a multivariable logistic regression model was calculated using stepwise forward–backward model selection based on the Akaike information criterion. All univariable risk indicators were included in the multivariable analysis.

Finally, the association of nine intraoperative risk indicators on the lowest median perioperative PhA was evaluated using the univariable and multivariable logistic regression model approach, as described above. The reference groups for categorical risk indicators with more than two classes were no angina pectoris, LVEF more than 50%, and CABG. Statistical analysis was performed, and plots were drawn using the R 3.4.3 statistical environment (http://www.R-project.org/, accessed on 11 August 2019). *p*-values of 0.05 or less were considered statistically significant.

## 3. Results

### 3.1. Patient Characteristics

In total, 227 patients were screened for eligibility. Of these, 22 patients were excluded. We approached 205 patients, 5 of which declined informed consent. After including 200 patients, 32 dropped out. Finally, 168 patients were included in the analysis (Figure 1).

The mean age of our patients was 65.9 ± 11.8, and 34% (*n* = 57) were females. CABG, valve, and combined procedures were performed in 17% (*n* = 28), 55% (*n* = 92), and 26% (*n* = 43) of patients, respectively. Other procedures were performed in 3% (*n* = 5) of patients, including one septal myectomy, one ventricular septal defect closure, and three procedures on the ascending aorta. The mean procedure time was 308 ± 86 min. An average SAPS 3 score of 40.6 ± 11.3 was found after admission to the ICU, and the median length of ICU stay was 3.1 ± 4.2 days (Table 1).

### 3.2. Phase Angle Measurements

The preoperative baseline PhA was 6.1° ± 1.9°. On the first POD, the PhA decreased significantly to 4.0° ± 5.2° with a mean difference of −2.1° (95% CI, −2.9 to −1.3°; *p* < 0.0001), reaching a nadir of 3.5° ± 2.5° on POD 2 with a mean difference of −2.6° (95% CI, −3.0 to −2.1°; *p* < 0.0001). The perioperative decline in PhA remained significant until the end of the observation period on POD 7 (4.1° ± 2.4°), and did not return to baseline values (Figure 2A).

### 3.3. Comparison of Low versus High Preoperative Phase Angle

Patients were split into two groups depending on the preoperative median PhA of 5.84° [IQR 4.9° to 6.9°]. Group PhA_low_ was defined as patients below or equal to the preoperative median PhA, and group PhA_high_ was defined as patients above the preoperative median PhA. Patients with a preoperative PhA_low_ remained significantly lower compared with patients with a preoperative PhA_high_ until the sixth POD; on POD 7, no difference was found (PhA_low_ 3.8° ± 2.6° vs. PhA_high_ 4.5 ± 2.0°; *p* = 0.1478. Patients with a preoperative PhA_low_ were significantly older (71.0 ± 9.1 years vs. 60.9 ± 12.0 years; *p* < 0.0001), frailer (3.1 ± 1.3 vs. 2.3 ± 1.1; *p* < 0.0001), and had lower serum albumin levels (38.3 ± 4.3 g/L vs. 40.2 ± 3.1 g/L; *p* = 0.0013) compared with PhA_high_. We found a significantly higher cumulative fluid balance on the day of surgery in the PhA_low_ group, 8388 ± 3168 mL vs. 7417 ± 2459 mL, as compared with in the PhA_high_ group, *p* = 0.0287. Additionally, patients with a PhA_low_ had a higher SOFA (7.7 ± 2.3 vs. 6.9 ± 2.0, *p* = 0.0185), SAPS 3 score (42.5 ± 12.7 vs. 38.8 ± 9.6, *p* = 0.0391) and a longer stay in the ICU (3.7 ± 4.5 days vs. 2.6 ± 3.8, *p* = 0.0182). Detailed information can be found in Table 1.

### 3.4. Preoperative Risk Indicators on Preoperative Phase Angle

Univariable analysis indicated significant associations between three out of thirteen risk indicators on higher preoperative PhA levels for age (OR 0.96; 95% CI 0.94 to 0.99; *p* = 0.0024), frailty scale (OR 0.71; 95% CI 0.57 to 0.88; *p* = 0.0018) and female gender (OR 0.41; 95% CI 0.23 to 0.73; *p* = 0.003). Multivariate analysis indicated a significant association of lower frailty scale (OR 0.77; 95% CI 0.61 to 0.98; *p* = 0.0344) and higher LVEF (OR 1.85; 95% CI 1.07 to 3.19; *p* = 0.028) on preoperative higher PhA (Table 2).

### 3.5. Intraoperative Risk Indicators on the Nadir Phase Angle

Univariable analysis indicated significant associations between three out of nine intraoperative risk indicators on the nadir PhA on POD 2: combined procedure (OR 0.27; 95% CI 0.08 to 0.90; *p* = 0.0356), longer surgery time (OR 0.99; 95% CI 0.99 to 0.99; *p* = 0.0405), and higher fluid balance on the day of surgery (OR 0.83; 95% CI 0.73 to 0.96; *p* = 0.011). Multivariate analysis indicated a significant association of combined procedure (OR 0.28; 95% CI 0.08 to 0.99; *p* = 0.05), longer CPB time (OR 0.99; 95% CI 0.98 to 0.99; *p* = 0.0344), and higher fluid balance on the day of surgery (OR 0.86; 95% CI 0.75 to 0.99; *p* = 0.0371) on the lower PhA on POD 2 (Table 3).

## 4. Discussion

In this large cohort study of cardiac surgical patients, we demonstrated that the PhA can not only be used as a parameter for the health and nutritional status of patients, but also as a non-invasive and easy-to-use bedside instrument to evaluate preoperative risk and possibly identify patients at risk of a more complicated perioperative course.

To the best of our knowledge, this is the first study assessing a daily PhA profile in cardiac surgical patients; we found that the PhA decreases dramatically by more than 40% compared with preoperative baseline values in all patients after cardiac surgery and never reaches preoperative levels again within seven days. This decrease was more pronounced in patients with low preoperative PhA values. Other studies have shown a decrease in PhA over time after cardiac surgery [16,23]; however, none have shown that the deterioration occurs immediately postoperatively.

When analyzing the dramatic postoperative decrease in PhA, we found combined procedures, longer CPB duration, and increased fluid resuscitation to be independent risk indicators. All these factors have been associated with a worse outcome after cardiac surgery [24,25,26,27], and indicate a more complicated intraoperative course.

PhA as a marker for cell health is also used to evaluate the extent of fluid redistribution between intracellular and extracellular fluid spaces [9,16,17,28]. In this regard, we observed that patients with a lower preoperative PhA had a significantly higher fluid balance on the day of surgery than patients with a higher preoperative PhA. This indicates that the cell membranes in these patients are more permeable; thus, a higher volume supply is required to maintain circulation.

It has been shown that volume overload in cardiac surgery patients can lead to a more severe postoperative course, in terms of congestion and associated organ dysfunction, such as acute kidney injury, and thus, increased morbidity and mortality [29,30,31]. Moreover, PhA is a parameter which is dependent on the fluid balance; it is possible that it may also describe congestion in patients with heart failure or reduced left ventricular function [17,28]. Nevertheless, the deterioration of PhA indicates the loss of integrity and quality of metabolic active cell mass, which means a reduction in functional reserves [9].

Patients with a comparatively lower preoperative PhA were older and frailer, needed more resuscitative fluids on the day of surgery, and had a significant longer postoperative ICU stay. As associated factors for a lower preoperative PhA, we found increased frailty and worse left ventricular function. This means that patients with lower PhA levels were in worse overall condition than patients with normal or higher PhA levels. This seems to be similar to a Brazilian study in which the authors found that patients’ preoperative PhA was inversely correlated with mechanical ventilation time, age, and EuroSCORE [23].

PhA can also be used as a marker for malnutrition [32]. Although the cardiac surgery population is very susceptible to malnutrition, it is rarely detected [18]. This malnutrition can be masked by increased BMI rates in cardiac surgical patients, which can also be caused by water retention following reduced cardiac function [33]. In our cohort, we found a mean BMI of 27.8 ± 4.9 kg/m^2^, and only two patients fulfilled the ESPEN criteria for malnutrition of a BMI lower than 18.5 kg/m^2^ [34]. Therefore, Ringaitiene and colleagues proposed the usage of PhA to detect the first nutritional alterations in cardiac surgical patients [18]. We observed a trend in gender difference for women being at risk for lower PhA. This can be a sign for malnutrition in females, who have a higher overall prevalence of malnutrition [35].

Serum albumin can also be used as a malnutrition marker; low levels are associated with adverse outcomes after cardiac surgery [36]. Although we found lower serum albumin levels in patients with lower preoperative PhA, the preoperative PhA was not influenced by the preoperative serum albumin levels in our cohort. Nevertheless, our observed serum albumin levels were higher than reported levels associated with cardiac cachexia or increased mortality, even in the cohort with low PhA [37].

There are several limitations to this study. First, the patients included in this study were all admitted for cardiac surgery; thus, they differ from the general population. Therefore, we have to mention the general limitations of BIA, which is not validated in an unhealthy cardiac surgery population and might therefore not be sufficiently accurate. Second, a single-center study such as this is limited in the interpretation of the results and not generalizable due to different institutional standards. Third, only one-third of patients were female in our study population. It is known that the PhA is lower in women than in men [38], but the gender distribution reflects the normal distribution in our cardiac surgical patients [39]. Moreover, we did not monitor hemodynamic parameters and vasopressor and/or inotropic use for this study. Intra- and postoperative fluid management, as well as vasopressors and/or inotropic use, was at the discretion of the attending consultant and not controlled by protocol. However, all procedures were performed by experienced cardiac anesthesia fellows supervised by senior cardiac anesthesiologists, who were all trained in echocardiography and invasive hemodynamic management. Lastly, we did not include blood loss in the fluid balance because we do not have consistent data on blood loss, and it is known that estimations of blood loss is frequently inaccurate and unreliable [40].

In conclusion, adding the BIA providing PhA into pre- and perioperative risk assessment may be beneficial in the field of cardiac surgery and can possibly positively influence the patient outcome. BIA is an easy-to-use and reliable bedside parameter to establish detailed risk profiles in cardiac surgical patients. A low preoperative PhA is associated with frailty and reduced LVEF and is predictive for a longer length of ICU stay after cardiac surgery.

## Figures and Tables

**Figure 1 nutrients-14-02491-f001:**
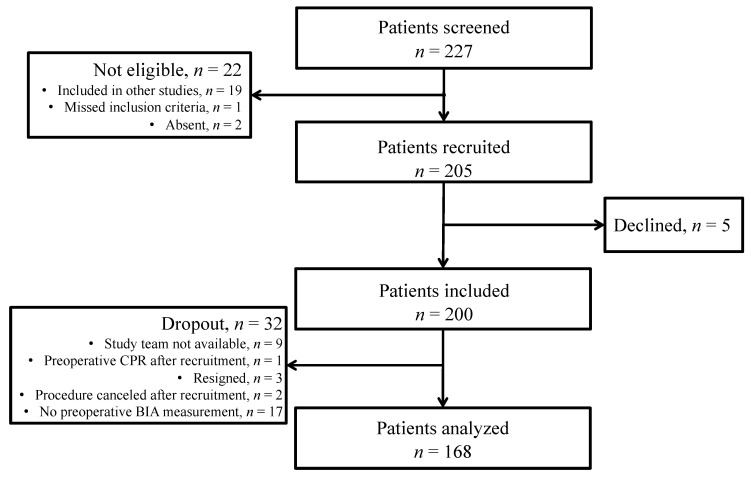
Selection and exclusion criteria for patients enrolled in the study.

**Figure 2 nutrients-14-02491-f002:**
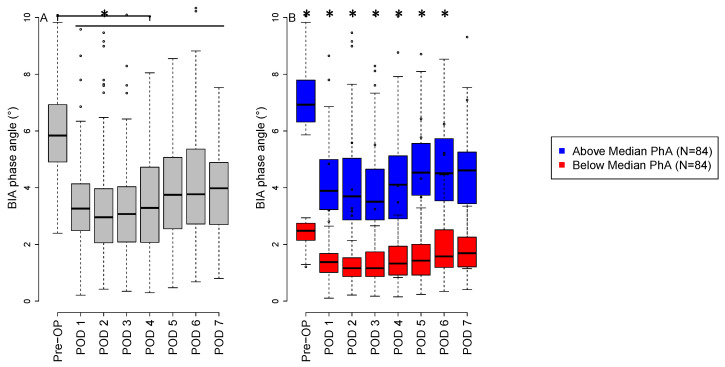
Perioperative phase angle. (**A**) The boxplots show the perioperative course of phase angle at different time points. (**B**) The boxplots show the difference between patients’ perioperative course of high (blue) vs. low (red) preoperative phase angles at different time points. Asterisks mark significant differences between the two groups at a *p*-value of less than 0.05. In the boxplots, the lower boundary of the box indicates the 25th percentile, a black line within the box marks the median, and the upper boundary of the box indicates the 75th percentile. Whiskers above and below the box indicate the 10th and 90th percentiles. Points above and below the whiskers indicate outliers outside the 10th and 90th percentiles. Abbreviations: BIA, bioelectrical impedance analysis; Pre-OP, preoperative; POD, postoperative day.

**Table 1 nutrients-14-02491-t001:** Demographic and surgical characteristics.

	All	Phase Angle_low_	Phase Angle_high_	*p*-Value
Male	111 (66.1)	42 (50)	69 (82.1)	0.0227
Female	57 (33.9)	42 (50)	15 (17.9)	
Age (yrs)	65.9 ± 11.8	71.0 ± 9.1	60.9 ± 12.0	<0.0001
BMI (kg/m^2^)	27.8 ± 4.9	27.9 ± 5.4	27.7 ± 4.5	0.8058
Frailty scale	2.7 ± 1.3	3.1 ± 1.3	2.3 ± 1.1	<0.0001
Albumin (g/L)	39.3 ± 3.9	38.3 ± 4.3	40.2 ± 3.1	0.0013
**Preoperative characteristics**				
Resistance	401.0 ± 80.1	419.2 ± 80.2	382.7 ± 76.0	0.0028
Reactance	42.0 ± 13.1	34.7 ± 9.3	49.2 ± 12.3	<0.0001
Phase angle	6.1 ± 1.9	4.7 ± 0.8	7.4 ± 1.7	<0.0001
**Comorbidities**				
Asthma	7 (4.2)	5 (6.0)	2 (2.4)	0.44
COPD	22 (13.1)	12 (14.3)	10 (11.9)	0.8191
NIDDM	26 (15.5)	13 (15.5)	13 (15.5)	1.0
IDDM	8 (4.8)	5 (6.0)	3 (3.6)	0.7171
Chronic kidney disease	14 (8.3)	8 (9.5)	6 (7.1)	0.7801
Cardiac decompensation	1 (0.6)	0 (0)	1 (1.2)	1.0
PAOD	11 (6.6)	8 (9.5)	3 (3.6)	
Atrial fibrillation	40 (23.8)	24 (28.6)	16 (19.5)	
Angina pectoris				
Absent	124 (73.8)	65 (77.4)	59 (70.2)	0.4028
Stable	41 (24.4)	17 (20.2)	24 (28.6)	
Unstable	3 (1.8)	2 (2.4)	1 (1.2)	
LVEF				
>50%	133 (79.2)	65 (77.4)	68 (81.0)	0.4501
30–50%	29 (17.3)	17 (20.2)	12 (14.3)	
<30%	6 (3.6)	2 (2.4)	4 (4.8)	
**Surgical characteristics**				
Procedure				
CABG	28 (16.7)	11 (13.1)	17 (20.2)	0.05506
Combined	43 (25.6)	25 (29.8)	18 (21.4)	
Valve	92 (54.8)	48 (57.1)	44 (52.4)	
Others	5 (3.0)	0 (0)	5 (6.0)	
Reoperation	21 (12.5)	13 (15.5)	8 (9.5)	0.3507
Anesthesia duration (min)	396 ± 92	399.4 ± 90.0	392.7 ± 93.9	0.6397
Surgery (min)	308 ± 86	310.4 ± 82.9	304.7 ± 90.2	0.6675
CPB (min)	149 ± 59	151.9 ± 54.7	145.1 ± 63.7	0.4554
AoCC (min)	96 ± 45	93.9 ± 43.4	98.9 ± 45.6	0.4722
Balance_intraoperative_ (mL)	4828 ± 2290	5145 ± 2556	4516 ± 1958	0.0765
PRBC (units)	0.7 ± 1.2	1.0 ± 1.4	0.4 ± 0.9	0.0014
Platelets (units)	0.2 ± 0.5	0.3 ± 0.6	0.1 ± 0.4	0.0470
Fresh frozen plasma (units)	0.1 ± 0.5	0.1 ± 0.5	0.1 ± 0.6	0.7659
Fibrinogen (g)	0.7 ± 1.2	0.9 ± 1.4	0.5 ± 0.9	0.0568
**Postoperative risk indicators**				
Fluid balance_day of surgery_ (mL)	7621 ± 2867	8388 ± 3168	7417 ± 2459	0.0287
SAPS 3	40.6 ± 11.3	42.5 ± 12.7	38.8 ± 9.6	0.0391
SOFA on ICU admission	7.3 ± 2.2	7.7 ± 2.3	6.9 ± 2.0	0.0185
Length of ICU stay (d)	3.1 ±4.2	3.7 ± 4.5	2.6 ± 3.8	0.0182

Values are presented as the number (*n*) and percentage (%) or mean (standard deviation). Abbreviations: AoCC, aortic cross-clamp; CABG, coronary artery bypass graft; COPD, chronic obstructive pulmonary disease; CPB, cardiopulmonary bypass; d, days; ICU, intensive care unit; IDDM, insulin-dependent diabetes mellitus; LVEF, left ventricular ejection fraction; NIDDM, non-insulin-dependent diabetes mellitus; PAOD, peripheral artery occlusive disease; PRBC, packed red blood cells; SAPS, simplified acute physiology score; SOFA, sepsis-related organ failure assessment score; yrs, years.

**Table 2 nutrients-14-02491-t002:** Logistic regression analysis of risk indicators influencing the preoperative phase angle.

	Univariable		Multivariable	
Risk Indicator	OR (95% CI)	*p*-Value	OR (95% CI)	*p*-Value
Age (yrs)	0.96 (0.94; 0.99)	0.0024	0.98 (0.95; 1.02)	0.0791
BMI (kg/m^2^)	0.97 (0.91; 1.02)	0.288		
Frailty scale	0.71 (0.57; 0.88)	0.0018	0.77 (0.61; 0.98)	0.0344
Female	0.41 (0.23; 0.73)	0.0030	0.56 (0.31; 1.01)	0.0558
Albumin (g/L)	1.07 (0.99; 1.15)	0.0843		
Asthma	0.52 (0.12; 2.15)	0.365		
COPD	0.65 (0.28; 1.52)	0.326		
NIDDM	0.93 (0.42; 2.05)	0.855		
IDDM	0.45 (0.12; 1.70)	0.239		
Chronic kidney disease	0.97 (0.34; 2.72)	0.949		
Cardiac decompensation	3.57 (0.09; 146.8)	0.503		
PAOD	0.43 (0.14; 1.36)	0.154		
Angina pectoris	1.05 (0.58; 1.88)	0.881		
LVEF	1.65 (0.94; 2.88)	0.0834	1.85 (1.07; 3.19)	0.0280

Values are presented as the odds ratio (95% confidence interval). For multivariate analysis, a backward–forward selection of univariate risk indicators was used. For angina pectoris and LVEF, the absence of angina pectoris or LVEF was used as a reference. Abbreviations: BMI, body mass index; COPD, chronic obstructive pulmonary disease; IDDM, insulin-dependent diabetes mellitus; LVEF, left ventricular ejection fraction; NIDDM, non-insulin-dependent diabetes mellitus; PAOD, peripheral artery occlusive disease; yrs, years.

**Table 3 nutrients-14-02491-t003:** Logistic regression analysis of intraoperative risk indicators influencing the nadir phase angle.

	Univariable		Multivariable	
Risk Indicator	OR (95% CI)	*p*-Value	OR (95% CI)	*p*-Value
CABG	1		1	
Valve procedure	0.96 (0.33; 2.79)	0.9397	0.79 (0.27; 2.33)	0.6714
Combined procedure	0.27 (0.08; 0.90)	0.0356	0.28 (0.08; 0.99)	0.0500
Other procedure	3.35 (0.33; 33.94)	0.3083	2.06 (0.21; 20.67)	0.5400
Reoperation	0.76 (0.24; 2.45)	0.649		
Surgery time (per min)	0.99 (0.99; 0.99)	0.0405		
CPB time (per min)	1.0 (0.98; 1.00)	0.118	0.99 (0.98; 0.99)	0.0344
AoCC time (per min)	1.00 (0.99; 1.01)	0.892	1.01 (1.00; 1.03)	0.0514
Fluid balance_day of surgery_ (per Liter)	0.83 (0.73; 0.96)	0.011	0.86 (0.75; 0.99)	0.0371

Values are presented as the odds ratio (95% confidence interval). For multivariate analysis, a backward–forward selection of univariate risk indicators was used. For surgical procedures, CABG was used as a reference. Abbreviations: AoCC, aortic cross-clamp; CABG, coronary artery bypass graft; CPB, cardiopulmonary bypass.

## Data Availability

Not applicable.

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
