# Peer review of "Preoperative Phase Angle as a Risk Indicator in Cardiac Surgery—A Prospective Observational Study"

_nutrients, 2022, doi:10.3390/nu14122491_

Round 1

Reviewer 1 Report

the study is very original and interesting  cause the investigated cohort of cardiac disease patients. I suggest to improve the quality of the english style. we can offer a native language medical writer  to "wash out" the grammar. 

I suggest to correct the right citation of the technology  as suggested into the attachment since there are many confusion in the bioimpedance manufacturer and models.  the BIA measurements are branded specific and it's mandatory being very specific in the citation and in the cross references .

I also suggest to cite this paper too into your references :

de Borba, Evandro Lucas, et al. "Phase angle of bioimpedance at 50 kHz is associated with cardiovascular diseases: systematic review and meta-analysis." European Journal of Clinical Nutrition (2022): 1-8.

Reviewer 2 Report

The authors measured PhA daily from preoperative to 7 days postoperatively in 168 elective cardiac surgical patients. The authors then analyzed the relationship between preoperative PhA levels in the subjects and various existing risk indicators and the postoperative course. As a result, low preoperative PhA levels were associated with the subject's age, weakness, more fluid replacement, and cardiac function. On the other hand, postoperative low PhA levels were associated with postoperative higher fluid balances and longer cardiopulmonary bypass time.

The use of PhA has gained attention as alternative to conventional error-prone calculation of body composition in disease. This study is an observational study, not an intervention study. However, this study analyzed the relationship between the PhA value before and after surgery in heart surgery patients, various risk factors, and the postoperative course, and is considered to have great clinical significance. No major problems were felt in the study design, evaluated indicators, or analysis method. However, I have a few questions.

The authors use cumulative fluid balances as an indicator to assess the relationship between PhA and fluid volume. However, we did not evaluate the association between PhA and indicators that more directly evaluate hemodynamics such as blood pressure and central venous pressure. Why is this?

The authors also mention PhA as a nutritional index in the discussion. However, the authors did not evaluate the indicators that directly evaluate the nutritional status of the target patients, such as serum albumin level and total serum protein level, in this study. Why is this?
